# Implementing the Semantic Units Framework using Named Graphs and R2RML in a Knowledge Graph for the Biodiversity Exploratories

Tarek Al Mustafa[1,2,*], Lars Vogt[3], Tina Heger[4,5,6] and Birgitta Koenig-Ries[1,2]

[1]*Friedrich Schiller University Jena, Fürstengraben 1, 07743 Jena, Germany*

[2]*German Centre for Integrative Biodiversity Research (iDiv) Halle-Jena-Leipzig, Puschstr. 4, 04103 Leipzig, Germany*

[3]*Leibniz Institute for the Analysis of Biodiversity Change, Museum of Nature Hamburg, Martin-Luther-King Platz 3, 20146 Hamburg, Germany*

[4]*Leibniz Institute of Freshwater Ecology and Inland Fisheries (IGB), Müggelseedamm 310, 12587 Berlin, Germany*

[5]*Freie Universität Berlin, Institute of Biology, Königin-Luise-Str. 1–3, 14195 Berlin, Germany*

[6]*Technical University of Munich, Germany; TUM School of Life Sciences, Restoration Ecology, Emil-Ramann-Str. 6, 85354 Freising, Germany*

## Abstract

Knowledge Graphs (KGs) can support many applications in the domains of ecology and biodiversity research, especially by facilitating synthesis efforts and the creation of FAIR datasets. Semantic units have been proposed to organize a KG into semantically meaningful subgraphs using named graphs, to simplify cognitive interoperability for users, tackle common KG modeling and knowledge representation challenges, and facilitate reasoning over diverse logical frameworks, resulting in FAIR and CLEAR KGs. However, no implementation beyond a small proof of concept and thus no evaluation of semantic units currently exists, which motivates our contributions: We construct a KG for publication and dataset metadata of the Biodiversity Exploratories, a platform for functional biodiversity research in three study regions across Germany, and contribute the first implementation of semantic units on that graph. We present relevant theoretical background on semantic units, detail our KG modeling methodology, and semantic unit implementation method using the R2RML mapping language and named graphs. Finally, we list the findings from a competency question evaluation that compares SPARQL queries with and without semantic units and present the potential of semantic unit visualization to simplify the visual representation of KG content. We conclude with lessons learned and avenues for future work based on our implementation.

## Keywords

Knowledge Graphs, Semantic Units, Named Graphs, FAIR Principles, R2RML, Biodiversity Exploratories

## 1. Introduction

*Knowledge graphs* (KGs) have received ever growing attention in recent decades and are used in many applications in industry and science [1]. Even though their usage is as wide-spread as it is, interacting with knowledge graphs is notoriously difficult, even for users with a background in technical fields

---

*KGCW'26: 7th International Workshop on Knowledge Graph Construction, May 10, 2025, Dubrovnik, HRV*
*Corresponding author.

✉ tarek.al_mustafa@idiv.de (T. Al Mustafa); l.vogt@leibniz-lib.de (L. Vogt); t.heger@tum.de (T. Heger)
🌐 https://www.fmi.uni-jena.de/en/10758/tarek-al-mustafa (T. Al Mustafa);
https://leibniz-lib.de/de/ueber-das-lib/mitarbeitende/lars-vogt.html (L. Vogt); https://www.tinaheger.de/ (T. Heger);
https://www.fmi.uni-jena.de/en/10780/koenig-ries (B. Koenig-Ries)
🔗 0000-0001-7793-4483 (T. Al Mustafa); 0000-0002-8280-0487 (L. Vogt); 0000-0002-5522-5632 (T. Heger); 0000-0002-2382-9722 (B. Koenig-Ries)

[2, 3, 4, 5]. We argue that this challenge must be overcome, as especially OWL-based KGs bear great potential to support applications in ecology and biodiversity research: Semantification, metadata enrichment, machine-actionability that enables reasoning applications, integration and synthesis support; the creation of *FAIR* (Findable, Accessible, Interoperable, and Reusable) [6] and *CLEAR* (Cognitively interoperable, semantically Linked, contextually Explorable, easily Accessible, and human-Readable and -interpretable) [7] datasets, and overall reduction of manual labor are among the significant advantages of utilizing KGs.

*Semantic Units* (SUs) – semantically meaningful, named subgraphs in a knowledge graph – have been proposed in the literature [8, 9, 10, 11] to support users in understanding a KG's contents. However, currently no implementation of semantic units exists beyond a small proof of concept [8], with which to evaluate their effectiveness, making our contribution twofold: First, we develop a KG that consists of publication and dataset metadata of the *Biodiversity Exploratories*[1] (BE), an Infrastructure Priority Programme (PP 1374) funded by the *German Research Foundation* (DFG), that presents a research platform for functional biodiversity research on selected research plots across Germany [12]. We formulate competency questions with domain experts that aim to cover a multitude of scenarios of questions users may have for the KG. Second, we implement semantic units on that KG and investigate how they can be queried to solve representative tasks, and how these queries complement conventional queries. Further, we explore SU visualization and present upsides of this approach. Finally, we list suggestions for the SU framework based on lessons learned.

## 2. Background

Prior work proposes a semantic unit framework in the ecology domain [11], in which FAIR and CLEAR ecological data and knowledge is motivated by the fundamental challenges faced in knowledge representation when attempting to create KGs as models of the real world. In ecology, these challenges appear along many dimensions: Results of studies depend on the context of the study. The nature and direction of correlational or causal relationships may change depending on the layers of abstraction it may be viewed from, change over time, or be influenced by known or unknown confounders [13]. Technical constraints of KGs lead to challenges in knowledge representation. KG's reliance on triples to represent binary relationships introduces complexity in modeling n-ary relationships. Attempts to solve these challenges using complex semantic modeling, while, at the same time, aiming for machine-actionability, introduce a common issue: further complexity is added to the way knowledge is represented in OWL-based KGs, ultimately leading to a decline in *cognitive interoperability*[2] for users.

A proposed solution incorporates named graphs [14, 15] – sets of triples that are assigned an IRI (Internationalized Resource Identifier) denoting the whole set. These IRIs enable referencing of groups of triples to make statements about them. This allows application of the semantic modularization principle [10] to represent knowledge contained in a knowledge graph using semantically meaningful modules (groups of triples). These modules maintain semantic coherence and cognitive interoperability as their content is meaningful to users, enable nesting of modules to express complex statements, allow for contextual navigation between linked modules, and allow logical heterogeneity by enabling the use of different logical formalisms for specific modules when necessary [11].

**Semantic Units.** Semantic Units organize a KG, enabling the application of the semantic modularization principle. They are designed to denote semantically meaningful information for users and are developed for a specific domain of application. Thus, the source of their content is the *knowledge* a KG is intended

---

[1]https://www.biodiversity-exploratories.de/en/

[2]*'Cognitive interoperability refers to a system's ability to support intuitive and efficient human interaction with data and metadata. It focuses on aligning the complexity of data structures (human-information interaction) and interfaces (human-computer interaction) within human cognitive capacities. Systems that support cognitive interoperability offer tools for exploration, trust building, and integration into workflows, making them accessible to database architects, data scientists, and domain experts alike.'* [11]

to convey to its users, partitioned into semantically meaningful units. Figure 1 shows the fundamental structure of all semantic units: The blue box at the bottom denotes the semantic content to be represented by a SU. This semantic content consists of one or more triples from the KG and is stored in its own named graph (*content-graph* in Fig. 1 for statement units - merging the content-graphs of all statement units results in the base KG without SUs) or is a collection of SU resources (see compound units below). Each SU can be identified by its own resource (blue box with purple border) which, in the case of a statement unit, is also the resource of its corresponding named graph. This resource instantiates a specific semantic unit class. Additional metadata properties can be appended to a semantic unit which is especially useful to capture provenance data, including a specification of the logical framework used to model the semantic unit's content.

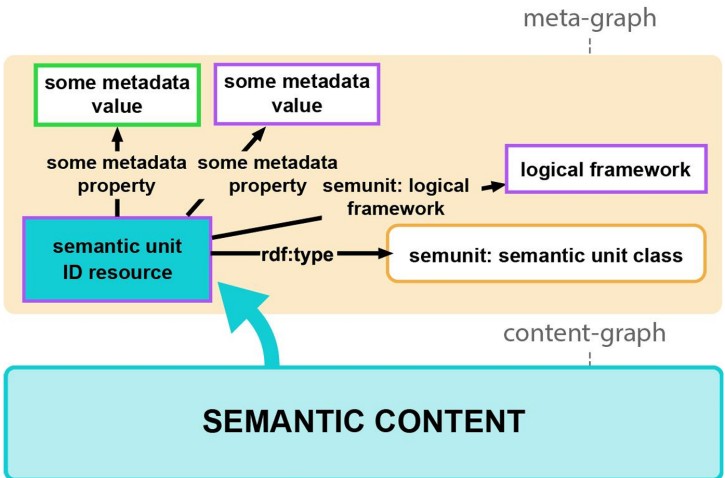

**Figure 1:** '**Basic structure of a semantic unit, implemented in an OWL/RDF framework.**' Semantic units contain two main parts: The blue rectangle at the bottom contains the semantic content of the semantic unit. This semantic content consists of one or more triples from the KG (*content-graph*). The top consists of new triples added in the SU framework that belong to the *meta-graph*. The *semantic unit ID resource* is an instance in the KG (the semantic unit), that identifies the semantic content (named graph). As a KG instance, it belongs to a specific semantic unit class. Further metadata might be attached to the SU and reference can be made to the logical framework used in the SU's modeling. Class resources are shown with yellow border, instance resources with purple border, and literals with green border.' Adapted from [11], Figure 1.

**Statement Units.** Statement units are the fundamental building blocks to represent propositions in the SU framework and contain the smallest amount of semantically meaningful content relevant to users, organized into named graphs [9, 10, 11]. For example, a statement unit can contain the proposition *'Person A is the first author of Publication P'*. Representing this proposition in an RDF/OWL based knowledge graph may require triples using the predicates *dcterms:creator and ro:hasRole*. A statement unit encloses relevant triples (one or many) to formulate this statement and assigns an identifier to the statement as a whole (i.e., the named graph IRI), denoting the author as the subject of the statement via *semunit:hasSemanticUnitSubject*. Further metadata properties can be appended to the statement unit, for example, to capture provenance information. By creating an OWL class for all statement units of the same statement type (example: *semunit:firstAuthorStatementUnit*), it becomes possible to query the KG for all statements of specific types (*return all first authorship statements*), and to query for all statements made about a subject (*return all semantic units Person A is subject of*). For each SU class, *SHACL (Shapes Constraint Language)*[3] shapes can be defined to validate their usage and guarantee consistent querying. For example, a SHACL shape can state that the subject of a *semunit:firstAuthorStatementUnit* must always be an instance of the class *foaf:person*.

**Compound Units.** Compound units [9, 10, 11] allow for the combination of multiple statement and compound units into coherent, identifiable, and reusable collections, that are each represented in the KG with their own resource instantiating a corresponding compound unit class to satisfy specific

---

[3]https://www.w3.org/TR/shacl/

information needs. Continuing the previous example, the creation of statement units to express propositions such as *'Person B is a co-author of Publication P'* of type *semunit:coAuthorStatementUnit* allows for the combination of both statement types into one compound unit. This compound unit contains all authorship statements about *Publication P*, (the semantic unit subject), and uses the predicate *semunit:hasAssociatedSemanticUnit* to reference relevant statements. Therefore, it becomes possible to retrieve all authorship statements across all publications or for a specific publication. Analogously to statement units, SHACL shapes can be defined for every compound unit class.

**Categories of Semantic Units.** Next to statement and compound units as the two fundamental SU types, the Framework provides a variety of additional SU classes to satisfy different kinds of information needs and to address common KG modeling challenges. The whole hierarchy of SU classes as described in [10] is shown in Fig. 2. We include this list of classes here to provide an overview of the extent of the SU Framework, and refer readers to [10] for further specification of SU classes and their usage scenarios.

**The Biodiversity Exploratories & BExIS.** The *Biodiversity Exploratories* (BE) are an Infrastructure Priority Programme (PP 1374) funded by the *German Research Foundation* (DFG), presenting a research platform for long-term functional biodiversity research on selected research plots across Germany [12]. The BE's main research objectives examine: *'how the form and intensities of land use affect biodiversity and ecosystem processes', 'how different components of biodiversity interact', and 'how different components of biodiversity influence ecosystem processes and ecosystem services'.*[4] *The Biodiversity Exploratories Information System (BExIS)*[5] is the BE instance of the research data management platform BEXIS2 [16]. It provides public access to more than 1500 datasets. To construct the KG for this work, we pull all publication and dataset metadata up to February of 2026 from the BExIS API for further processing.

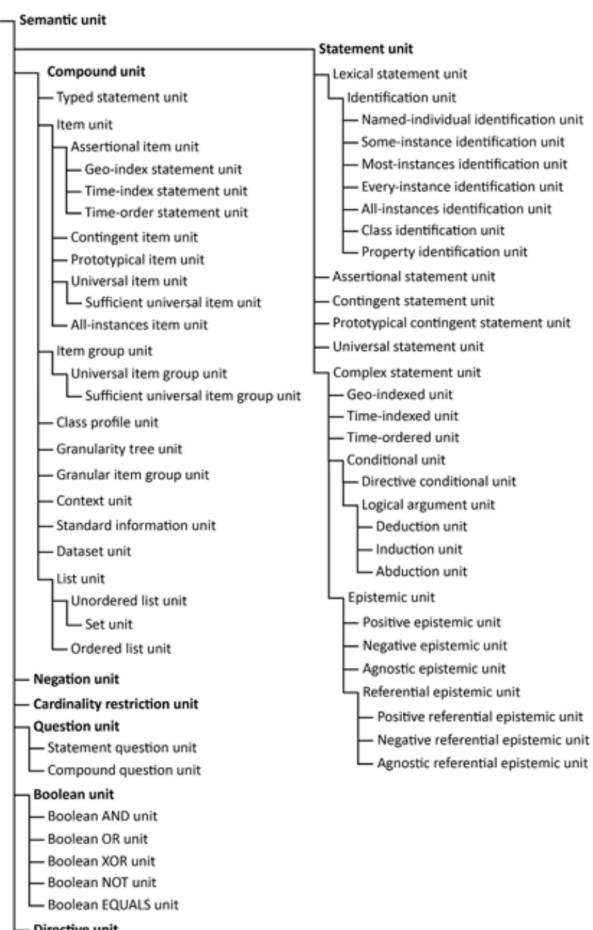

**Figure 2:** 'Classification of different categories of semantic units.' Adapted from [10], Figure 21.

# 3. Related Work

Core principles behind the Semantic Units Framework are the modularization of KG contents and the possibility to make statements about statements in KGs. Solutions for these issues have been developed by the research community in the past, and here we aim to discuss what value the SU Framework adds beyond the application and combination of already existing approaches.

**Named Graphs.** Named graphs have been proposed as a tool to attach provenance and other metadata to KG triples [17, 18, 19]. They are syntax-/representation-level tools: sets of RDF triples identified by a graph IRI, to which metadata can be attached. SUs contribute conceptual and methodological

---

[4]https://www.biodiversity-exploratories.de/en/about-us/research-objectives-and-background/
[5]https://www.bexis.uni-jena.de/

abstractions: the semantically meaningful content, metadata, and its intended interpretation are treated as a single epistemic unit of meaning and use. Therefore, they are resources that can be typed and linked to SHACL shapes as pre-defined or custom classes, validated, cited, and explored.

Named graphs can implement SUs, but they do not define what constitutes a unit, its typing, how metadata representation is standardized (meta-graph conventions), how units are composed into bundles to represent complex information needs, or how unit-level operations (validation, exploration) are performed. Therefore, the contribution of the SU Framework is not as a tool for implementation but rather a technology-agnostic formalization for knowledge and property graphs that enables uniform KG construction and validation workflows and the alignment of KG content across graphs over shared SU classes. While statement units can be modeled as named graph, compound units represent collections of SUs that do not require named graphs for being modeled. They reference the IRIs of the semantic units (statement and/or compound units) that belong to their collection and are typed themselves. This enables to organize a knowledge graph into different levels and layers of information granularity.

**Reification, RDF-star, and Singleton Properties.** Besides named graphs, Reification[6], RDF-star[7], and singleton properties [20] have been raised as tools to make statements about statements. Reification allows the attachment of metadata to a triple by creating a new RDF resource that identifies the triple itself, the contents of which are attached to that resource via three properties denoting the triple's subject, predicate, and object. RDF-star as an extension of RDF, aims to provide an alternative to reification by allowing subjects and objects of a triple to also be triples instead of standard resources using *quoted triples*[8]. Finally, singleton properties have been proposed as a further alternative to reification that is still RDF compliant: 'A singleton property is a property instance representing one specific relationship between two particular entities under one specific context. Singleton properties can be viewed as instances of generic properties whose extensions contain a set of entity pairs' [20].

There has been previous discussion about which of these tools is suited best for attaching metadata to KG triples [21, 22, 23, 24]. As for the SU Framework, we chose to implement named graphs as they are the simplest and most interoperable implementation choice in RDF while satisfying three main requirements: (1) provide an identifier for a 'container' of triples, (2) attach metadata to that container and (3) allow statement units to contain multiple triples to express n-ary relationships that are semantically significant for users. Reification and RDF-star can (with workarounds) satisfy (1) and (2), however, they do not natively support (3). Thus, a multi-triple statement unit would require an explicit grouping layer with nodes that assign membership links to multiple reified or RDF-star quoted triples. This would introduce join-heavy querying, and more difficult subgraph retrieval and graph exploration, while also introducing more triples to the graph. Singleton properties are not suitable for SU implementation as they focus on individual relations rather than groups of multiple triples, again requiring an additional grouping layer in the graph.

## 4. Method

**Modeling Approach.** We combine multiple best practices in our KG creation approach. We set up an interdisciplinary core team, including besides ontology engineers a domain expert (Tina Heger), as well as the creator of the semantic units framework (Lars Vogt). Further ecologists, the BExIS team and experienced ontology engineers were included in live modeling sessions and feedback rounds. The design was driven by competency questions derived from domain experts. We also reuse ontology classes and properties whenever possible resulting in 76/80 reused terms. Below, we list the resulting competency questions and show excerpts of the KG schema. For details on the entire KG schema, terms used, and our modeling decisions, we refer readers to [25].

---

[6]https://www.w3.org/TR/rdf11-mt/#reification
[7]https://w3c.github.io/rdf-star/cg-spec/2021-12-17.html
[8]https://w3c.github.io/rdf-star/cg-spec/editors_draft.html#dfn-quoted

**Competency Questions**

- **CQ1:** Who authored publication *P*/dataset *D*; were they first or co-authors?
- **CQ2:** What are the link types to and from publication *P*/dataset *D*?
- **CQ3:** For publication *P*/dataset *D*, what are the plot levels of investigated plots, and what exploratory or multiple exploratories do those plots cover?
- **CQ4:** What are all semantic units that have publication *P*/dataset *D* as subject, or that are associated with it?
- **CQ5:** List all publications that belong to the infrastructure *Instrumentation/Remote sensing* that investigate pollination of plants on grassland plots. What are the datasets they reference and what projects do those belong to?

**Knowledge Graph Schema.** Fig. 3 provides an excerpt of the KG schema: The most fundamental design decision was to use the *Basic Formal Ontology (BFO)* [26] and BFO compliant vocabularies (*Ontology for Biomedical Investigations (OBI)* [27], *the Environment Ontology (ENVO)* [28], *Information Artifact Ontology (IAO)* [29], *OBO Relations Ontology (RO)* [30], etc.). In this framework we reuse two existing schemata to represent specific types of knowledge: The *OpenCitations Data Model* [31] allows us to represent documents, their creators, parts, citations, publishing information, and more. Second, the source data can be mapped easily to the OBI schema for scientific investigations, allowing us to represent study design, data and sample collection and processing information, and represent datasets of the BE. To represent the research design of the BE[9], we model infrastructures, projects, exploratories, and different plot levels and experiments within them. We also create forest and grassland environments in relation to the plots, and devise a schema to relate plots, collections of plots within a publication, environments, and taxa, qualities, and ecosystem processes investigated in publications to each other.

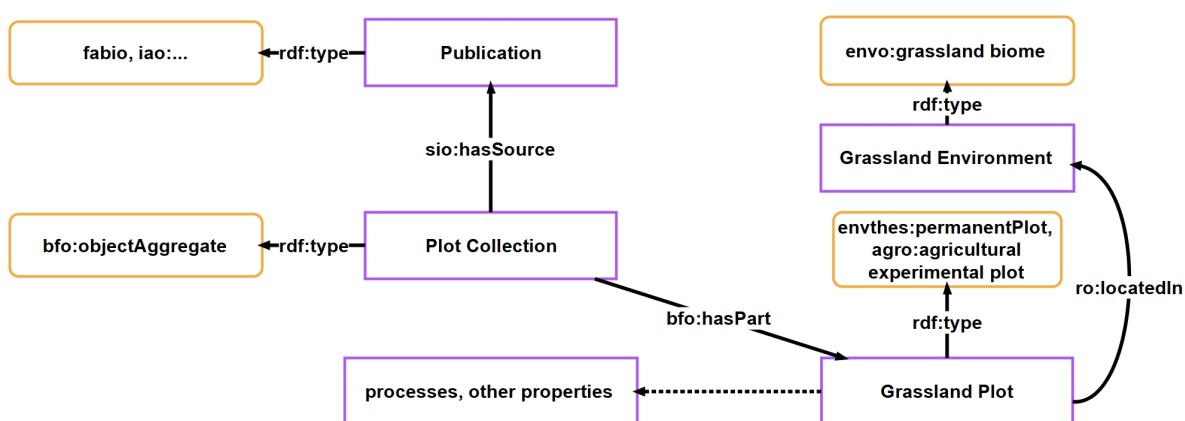

**Figure 3:** Simplified schema of instantiated plot collections (one collection of plots per publication), how they connect to their source publications, what plots they contain, and how processes and other properties are connected to those plots. Purple rectangles are instances and yellow-bordered rectangles are classes.

**Modeling Challenges.** We encountered several modeling challenges in this case study that we believe noteworthy, as they are likely to appear in other KG creation workflows:

- **Semi-structured input data.** BE metadata schemata introduce structure for some categories (definitions, descriptions, data type specifications, selection options), but also allow free-text answers for many categories reglated to experiment design. This leads to additional modeling and processing effort.
- **Metadata schemata for user convenience.** The BE require that researchers supply publication and dataset metadata to BExIS themselves. This has many advantages, but also introduces semantic ambiguities to the metadata forms to make them more accessible. For example, the category *Biotic Data Type* allows the answers *none*, *single species*, *multiple species*, *genetic*, and *total*

---

[9]https://www.biodiversity-exploratories.de/en/about-us/research-design/

*abundance* which mixes distinct dimensions (presence/absence, number of taxa, data modality, and aggregation level).

- **Missing concept definitions.** While most metadata categories have a general description of what to input, many descriptions lack clarity and do not define concepts properly. For example, the term *'genetic'* has ambiguous usage in the metadata, which made ontology linking difficult. We also assume that this ambiguity makes it more difficult for researchers to properly fill metadata categories.
- **Schematic inconsistencies.** In some cases, the metadata schemata for publications and datasets contain different categories, or allow different answers or supply different selection options for categories with the same name. For example, dataset metadata captures the exact number of plots per plot type and publication metadata only allows the options *'few'* or *'all'*.
- **Anonymous plots.** Unfortunately, the inconsistencies introduced above led to a significant issue: The metadata does not contain information about concrete research plots. Therefore, we can not make detailed deductions about what study took place on which plots. This problem also made it significantly more difficult to represent plot information in the knowledge graph.
- **Open World Assumption (OWA), negations, modeling complete information.** BE metadata contains *complete* information. For example, if metadata for a publication states that the study did not take place in forests, then the absence of triples relating to forests in the KG is intended. However, due to the OWA, it would follow logically that the publication might still have investigated forests, but the information was not provided, since lack of evidence does not equal evidence of lack. The representation of negations and uniting complete information with the OWA continue to be relevant questions in the domain. Unfortunately, we were not able to contribute to these questions further due to time constraints.

## 5. Implementation

We choose the *R2RML*[10] mapping language and processing engine[11] to create the KG's A-Box by mapping schema terms to tables stored in a *PostgreSQL*[12] database, as this approach has valuable advantages over other mapping strategies: First, the R2RML mapping language is highly expressive and allows to declare multiple conditionals in the mappings, that are difficult or impossible using other approaches (as we are using a relational database as the data source, R2RML even allows for results of SQL queries to be mapped). Second, R2RML makes virtualized knowledge graphs possible that allow SPARQL queries to be executed on a relational database by translating SPARQL to SQL using written mappings (we refer to ontop[13], a system that implements this), which allows formulation of combined queries over several knowledge graphs and relational databases. Third, R2RML allows to declare mappings for named subgraphs, which is required for the implementation of semantic units.

With R2RML, we are able to generate *TriG*[14] output, a *turtle*[15] (ttl) extension allowing for *graph maps* – mappings to declare named subgraphs inside of the main graph – and to declare which graphs specific triples belong to. Listing 1 illustrates an example of how a SU can be built using graph maps. The listing depicts two mapping entities, denoted by the notation <#...> in lines 1 and 28. The first mapping specifies the content-graph of a *creator statement unit* and the second its meta-graph. Each mapping declares a logical table (lines 3 and 30). These depict the table in the relational database the data originates from. Afterwards, a subject map is declared (lines 5 and 31). This assigns ontology classes to mapping entities. In lines 25 and 26, a graph map statement is declared resulting in the creation of a new subgraph with the IRI: *http://example.com/base/semunit/creatorStatementUnit/Publication_{id}*. This creates a named

---

[10]https://www.w3.org/TR/r2rml/
[11]https://github.com/chrdebru/r2rml
[12]https://www.postgresql.org/
[13]https://ontop-vkg.org/
[14]https://www.w3.org/TR/trig/
[15]https://www.w3.org/TR/turtle/

graph for each entry in the {id} column of the logical table and assigns the triple generated by the predicate object map (lines 19-26) to that named graph. This is how the *creator statement unit* is filled with triples from the content-graph. Below the graph map statement, object and data properties are declared using predicate object maps. For example, the semantic unit has two type assignments (lines 34 and 37) and a *semanticUnitSubject* property that connects it to the publication (line 41).

```
 1  <#Publication_id>
 2    a rr:TriplesMap ;
 3    rr:logicalTable [ rr:tableName "be_publication_metadata"] ;
 4
 5    rr:subjectMap [
 6        rr:template "http://example.com/base/Publication_{id}" ;
 7        rr:class iao:publication ;
 8        rr:termType rr:IRI;] ;
 9
10    rr:predicateObjectMap [
11    rr:predicate dcterms:creator ;
12    rr:objectMap [
13     rr:template "http://example.com/base/Author/{firstauthor_hash}" ;
14     rr:termType rr:IRI ;
15     ] ;
16     rr:graphMap [
17         rr:template "http://example.com/base/semunit/creatorStatementUnit/
                Publication_{id}" ;] ;] ;
18
19    rr:predicateObjectMap [
20    rr:predicate dcterms:creator ;
21    rr:objectMap [
22     rr:template "http://example.com/base/Author/{co_authors_hash}" ;
23     rr:termType rr:IRI ;
24     ] ;
25     rr:graphMap [
26         rr:template "http://example.com/base/semunit/creatorStatementUnit/
                Publication_{id}" ;] ; ] .
27
28  <#creatorStatementUnit_Publication>
29   a rr:TriplesMap ;
30   rr:logicalTable [ rr:tableName "be_publication_metadata" ] ;
31   rr:subjectMap [
32    rr:template "http://example.com/base/semunit/creatorStatementUnit/Publication_{id
          }" ;] ;
33   rr:predicateObjectMap [
34     rr:predicate rdf:type ;
35     rr:objectMap [ rr:constant semunit:creatorStatementUnit] ;] ;
36   rr:predicateObjectMap [
37     rr:predicate rdf:type ;
38     rr:objectMap [ rr:constant semunit:assertionalStatementUnit] ;] ;
39
40   rr:predicateObjectMap [
41     rr:predicate ex:semanticUnitSubject ;
42     rr:objectMap [ rr:parentTriplesMap <#Publication_id> ] ;] .
```

Listing 1: Excerpt of an R2RML mapping depicting mappings for publications and statement units depicting their creators.

Further, we present a minimal example of a SHACL shape for schema validation on the KG (Listing 2). We refer to the same example as in Listing 1. Lines 1 to 3 declare an entity that is a node shape, specifying that it is a SHACL shape for the class *iao:publication*. In lines 4 to 7, the shape validates that

there is at least one *dcterms:creator* relation connected to every publication.

The second SHACL shape begins in line 9, specifying that it targets entities of type *semunit:creatorStatementUnit*. This shape validates that every *creatorStatementUnit* also has the type *semunit:assertionalStatementUnit* (lines 12-15) and that every *creatorStatementUnit* must have exactly one *semunit:semanticUnitSubject* of type *iao:publication* (lines 16-21).

```
 1  ex:PublicationShape
 2     a sh:NodeShape ;
 3     sh:targetClass iao:publication ;
 4     sh:property [
 5        sh:path dcterms:creator ;
 6        sh:minCount 1 ;
 7        sh:message "Every publication must have at least one creator." ;] .
 8
 9  ex:CreatorStatementUnitShape
10     a sh:NodeShape ;
11     sh:targetClass semunit:creatorStatementUnit ;
12     sh:property [
13        sh:path rdf:type ;
14        sh:hasValue semunit:assertionalStatementUnit ;
15        sh:message "Creator statement units must also be typed as semunit:
               assertionalStatementUnit." ;] ;
16     sh:property [
17        sh:path semunit:semanticUnitSubject ;
18        sh:minCount 1 ;
19        sh:maxCount 1 ;
20        sh:class iao:publication ;
21        sh:message "A creator statement unit must have exactly one semanticUnitSubject,
               which must be a publication." ;] .
```

Listing 2: SHACL shapes to validate publication and creator statement unit triples.

## 6. Evaluation and Discussion

**Results, Reproducibility, and Availability.** The implemented mappings consist of ~ 11000 lines total, and are used to turn the contents of three PostgreSQL database tables into RDF files. The database tables consist of SQL exports from Ontotext Refine projects that contain raw data from the BExIS API, and hundreds of transformation steps, for which provenance data is saved in the project files. We upload the RDF files to the graph store *Ontotext GraphDB*[16]. The final KG contains 502,179 triples in total. We provide all files in a GitHub repository[17]. The KG is also published under a stable DOI.[18]

**SPARQL Query Patterns.** We present two query patterns that are useful for querying semantic units: a simple query listing all triples contained in a specific statement unit(Listing 3) and a query pattern returning all *associated* semantic units for a given compound unit (Listing 4). Depending on a user's information need, this query can be used for graph exploration by showing only associated SUs, or by also returning their content-graphs.

```
 1  SELECT ?s ?p ?o
 2  WHERE {
 3    GRAPH <StatementUnitIRI> { ?s ?p ?o }}
```

Listing 3: SPARQL query to retrieve all triples contained in a specific statement unit.

[16]https://www.ontotext.com/products/graphdb/
[17]https://github.com/fusion-jena/KG-for-Biodiversity-Exploratories-Metadata
[18]https://doi.org/10.71615/bexis.32235

```
1 SELECT * where {
2     <CompoundUnitIRI> semunit:hasAssociatedSemanticUnit* ?SU .
3   GRAPH ?SU { ?s ?p ?o . } }
```

Listing 4: SPARQL query to return *associated semantic units* and their content-graphs for a given compound unit.

**Evaluation using Competency Questions.** We compare two query approaches: First, conventional queries over the content-graph without reference to semantic units, and second, queries that include semantic units whenever possible. Additionally, competency questions can be split into two types: The first require queries to return information about a known resource (star shaped queries), and the second are answered by returning a list of entities that fit a specific description (set retrieval). All competency questions can be answered using SPARQL queries on the developed knowledge graph, however, we were unable to conduct a comprehensive user evaluation. Below, we discuss differences in SPARQL queries with and without semantic units.

**Differences in Query Approaches.** We present conventional and semantic unit SPARQL queries to answer the first competency question in Listings 5 and 6. For star shaped queries, we observe that the SU query is less complex. For the conventional query, we require the IRI of the first and co-author roles for a specific publication to return its authors. Querying for the same information using a the *authors and roles compound unit*, we retrieve authors and roles using the second query shape (Listing 4) by querying over *associated statement units* whose content-graphs contain triples that fit the *?role ro:roleOf ?person* pattern. We then retrieve author labels from outside the GRAPH pattern (alternatively, labels could be included in the statement unit's content-graph).

```
1 SELECT ?person ?label ?role WHERE {
2   ?person ro:hasRole ?role ;
3         rdfs:label ?label .
4   VALUES ?role {
5     <http://example.com/base/Role/FirstAuthor_31709>
6     <http://example.com/base/Role/CoAuthor_31709>} }
7 ORDER BY ?role ?label
```

Listing 5: SPARQL query to answer *Who authored the publication **P**? and were they first or co-author?* without using semantic units.

```
1 SELECT ?person ?label ?role WHERE {
2     <http://example.com/base/semunit/authorsAndRolesCompoundUnit/Publication_31709>
3         semunit:hasAssociatedSemanticUnit* ?su .
4   GRAPH ?su { ?role ro:roleOf ?person . }
5     ?person rdfs:label ?label .}
```

Listing 6: SPARQL query to answer *Who authored the publication **P**? and were they first or co-author?* utilizing semantic units.

For complex set retrieval queries (CQ5), we notice the following: The conventional query over the content-graph is able to retrieve relevant entities by traversing the KG, using two OPTIONAL clauses to capture incoming and outgoing links between datasets and publications. Formulating this query requires users to understand and traverse the KG's schema. To answer the same competency question using a compound unit (Listing 7), the query pattern from Listing 4 is applied repeatedly: The query searches *associated statement units* whose content-graphs contain the required information (lines 7, 11, 15, 19, 28, and 36). The set retrieval filters are applied in lines 23-26. Formulating this query still requires knowledge of the KG's schema, since properties must be referenced directly in the named graphs. However, if users execute the graph exploration query in Listing 4 before writing the set retrieval query, they can reuse the list of statement units and their content-graphs.

```
 1  SELECT * WHERE {
 2    ?compound a semunit:
           InfrastructureProcessAndServiceEnvironmentPublicationLinkProjectsCompoundUnit ;
 3          ex:semanticUnitSubject ?publication .
 4    ?publication obi:hasPart ?titleEntity .
 5    ?titleEntity a iao:documentTitle ;
 6          dcterms:description ?title .
 7    ?compound semunit:hasAssociatedSemanticUnit* ?suSource .
 8    GRAPH ?suSource {
 9      ?plotcollection dcterms:source ?publication .
10    }
11    ?compound semunit:hasAssociatedSemanticUnit* ?suPlotPart .
12    GRAPH ?suPlotPart {
13      ?plots obi:partOf ?plotcollection .
14    }
15    ?compound semunit:hasAssociatedSemanticUnit* ?suLoc .
16    GRAPH ?suLoc {
17      ?plots ro:locatedIn ?env .
18    }
19    ?compound semunit:hasAssociatedSemanticUnit* ?suHasPart .
20    GRAPH ?suHasPart {
21      ?process ro:hasParticipant ?plots .
22    }
23    ?publication obi:partOf <http://example.com/base/
           BE_Infrastructure_Instrumentation_RemoteSensing> .
24    ?env a envo:grasslandBiome .
25    ?plots obi:hasPart <http://www.gbif.org/species/6> .
26    ?process a <http://vocabs.lter-europe.net/EnvThes/21417> .
27    OPTIONAL {
28      ?compound semunit:hasAssociatedSemanticUnit* ?suTo .
29      GRAPH ?suTo {
30        ?projectto obi:hasSpecifiedOutput ?datasetto .
31      }
32      ?publication ?linkpropto ?datasetto .
33      ?projectto rdfs:label ?labelprojectto .
34    }
35    OPTIONAL {
36      ?compound semunit:hasAssociatedSemanticUnit* ?suFrom .
37      GRAPH ?suFrom {
38        ?projectfrom obi:hasSpecifiedOutput ?datasetfrom .
39      }
40      ?datasetfrom ?linkpropfrom ?publication .
41      ?projectfrom rdfs:label ?labelprojectfrom .
42    }
43  }
```

Listing 7: SPARQL query to answer CQ5 that queries over semantic units.

**Semantic Unit Visualization**. We are able to visualize the contents of specific semantic units using CONSTRUCT queries (c.f. Listing 8). Fig. 4 shows the result of such a query for compound units designed to answer the fifth competency question. This subgraph contains various information across the KG relating to a specific publication: The central node in the subgraph denotes the publication itself (light blue). It belongs to two BE infrastructures (light green), is connected to 9 datasets (pink), and is the source for a collection of plots relevant to a study investigating productivity of plants in grassland environments. Connected to each dataset is the project it is a specified output of (light green): Five datasets belong to the core project *Botany*, two to *SEBAS*, and two to the core project *Instrumentation and remote sensing*.

```
1  CONSTRUCT { ?s ?p ?o } WHERE {
2      <CompoundUnitIRI> semunit:hasAssociatedSemanticUnit* ?SU .
3    GRAPH ?SU { ?s ?p ?o . } }
```

Listing 8: Listing of a SPARQL CONSTRUCT query that returns the contents of all statement units associated with a specific compound unit.

Using this approach, it becomes possible to visualize complex relationships at considerable depth while only showing necessary information. Additionally, these visualizations can be saved as graph *snapshots* within GraphDB and shared between users. Snapshots could be utilized further in user interfaces to give users a better grasp of KG contents. Additionally, we note that the CONSTRUCT query used to generate this visualization can be formulated much simpler than one over the content-graph only, since the compound unit can be utilized.

**Lessons Learned and Future Work.** Finally, we list open questions for further discussion and avenues for future work that we identified in this work:

- Natural language to SPARQL translation and KG-based RAG approaches are promising research directions to enable KG question answering systems that lift the burden of query formulation from users. The semantic modularization that SUs contribute to KGs could be leveraged further by these systems, as current research already investigates schema-level KG features for question answering [32, 33, 34, 35].
- Including the generation of natural language labels/descriptions for the semantic content of statement and compound units in KG mappings could support both UIs and enable embedding based search on KGs. These *textual representations* of SU content are discussed further in [9, 36, 37].
- An aspect described in detail in [7] is enhanced *graph explorability* though the structure introduced in the SU-layer of the KG. When searching for specific information in a KG, or in an attempt to

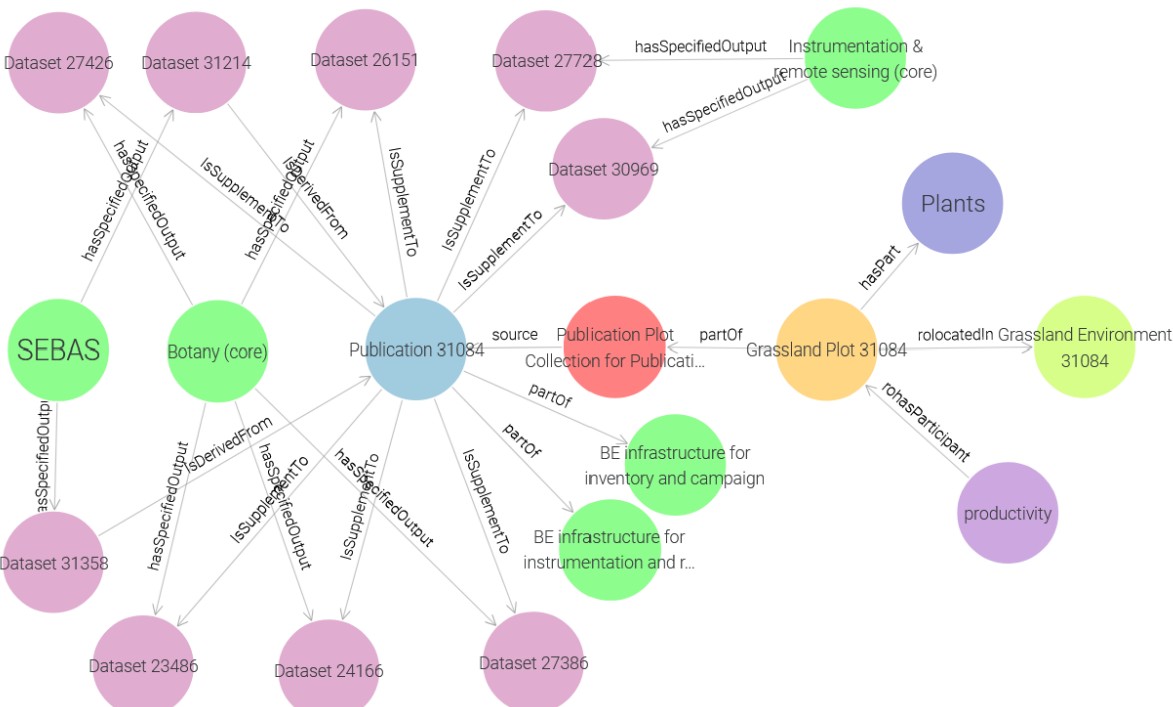

**Figure 4:** Visualization of the compound unit developed for the fifth competency question (light blue: publication 31084, pink: linked datasets and link types, light green: projects that have datasets as their specified output, remaining nodes to the right: plot collection for the publication and study contents)

better understand a KG's contents, users may employ querying, inspect specific KG resources and triples they appear in, or click through a visualization of the KG. This graph exploration behavior, comparable to navigational querying in web search [38], could thus be supported by the implementation of SUs in KGs, and should be evaluated in future work.

- Through our implementation workflow, it becomes clear that tooling support is necessary for wider adoption of SUs since hand-written mappings are prone to errors and time consuming.
- As listed in Fig. 2 in the background section, [9] and [10] provide a list of SU classes. Applying SUs and their types depends on the input data and domain of application and as such, no universal instructions for SU application can be declared. However, we may be able to identify sets of common use cases and provide a library for their reuse. For example, semantic units for the representation of scientific texts (e.g., the Open Citations Data Model used here), or semantic units for the OBI schema for scientific observations[19] are applicable across domains.

## 7. Conclusion

In this work, we constructed a knowledge graph for publication and dataset metadata of the Biodiversity Exploratories and contributed the first RDF/OWL-based implementation of the semantic units framework on that graph. Together with domain experts, we formulated competency questions and integrated them in the semantic modeling process, resulting in a KG of high semantic precision. We further demonstrated how the R2RML mapping language can be leveraged for SU implementation. Our evaluation showed that all competency questions could be answered both via traditional SPARQL queries and via queries over semantic units, but also revealed noteworthy differences between star and set retrieval queries. Further findings suggest that visualizations of SUs have great potential to present a KG's contents in a more cognitively interoperable manner for users. Finally, from the lessons learned in this implementation, we list open questions and identify avenues for future work, such as natural language descriptions of semantic unit content for UI applications and text search over the KG, use of semantic units for question answering and RAG systems, the utility of new tooling for semantic unit implementation, and the need for further evaluation of the framework.

## Acknowledgments

This work is based on data obtained within the DFG Priority Program 1374 'Biodiversity Exploratories' [12]. This research was supported by a Flexpool grant of the German Centre for Integrative Biodiversity Research Halle-Jena-Leipzig – iDiv. We thank the interdisciplinary resident group *Mapping Evidence to Theory in Ecology: Addressing the Challenges of Generalization and Causality* at the Center for Interdisciplinary Research (*ZiF: Zentrum für interdisziplinäre Forschung*)[20] at Bielefeld University that now continues as EcoWeaver[21].

**Data availability statement.** This work is based on the metadata of all datasets and publications stored in the Biodiversity Exploratories Information System (BExIS, https://doi.org/10.17616/R32P9Q) of the Biodiversity Exploratories program (DFG Priority Program 1374). The resulting dataset generated from these sources is listed in the References section as [39].

**Declaration on Generative AI.** The authors used generative AI (ChatGPT) to paraphrase and reword, improve writing style, and for grammar and spelling checks. After using the tool, the authors reviewed and edited the content as needed and take full responsibility for the publication's content.

---

[19]https://obi-ontology.org/docs/data-intro/
[20]https://www.uni-bielefeld.de/einrichtungen/zif/
[21]https://ecoweaver.hi-knowledge.org/

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
