# OpenReview forum: "Implementing the Semantic Units Framework using Named Graphs and R2RML in a Knowledge Graph for the Biodiversity Exploratories"
_eswc-conferences.org/ESWC/2026/Workshop/KGCW — KGCW 2026_

### Official Review · ~Jakub_Duchateau1 · 2026-03-31
**Nice start for Semantic Units, but doubt of the fairness of evaluation and questions remains.**

**Rating:** 6
**Confidence:** 3

**Review:**

This manuscript presents a practical implementation of a KG using the Semantic Units (SU) framework, applied to publication and dataset metadata from biodiversity research. Thank you for making me discover CLEAR; that was an interesting read. R2RML is used to construct the KG, and it uses the graph maps feature of R2RML. The authors mention some classical challenges of KGC: data quality, semi-structure data, semantic inconsistency, and the open-world assumption, but it seems they accommodate those. Then, the authors attempt to evaluate their KG modelisation against a set of competency questions developed with domain experts. Then proceed with comparing SPARQL query complexity (with and without using the Semantic Unit) on the constructed graph.

Overall, the manuscript presents a solid engineering contribution to the KGC Workshop by delivering the first documented real-world implementation of the Semantic Units framework. However, the paper's evaluation methodology needs refinement. The SPARQL query comparison appears unbalanced, as conventional queries are evaluated against a schema specifically optimised for SU, potentially inflating their perceived complexity. Additionally, discussions regarding scalability, query performance, and a comprehensive user evaluation are currently missing. Consequently, while the engineering effort is commendable, as a reader, I was left feeling that I have not learnt much about the actual practical trade-offs, operational bottlenecks, or performance implications of adopting the Semantic Units framework in a production environment.

## Strengths

- Follows principles such as FAIR and CLEAR to apply them to a real-world domain;
  - We have a reminder about cognitive interoperability, but it is left for future work
  - Novelty in implementation: this is the start of real-world implementation of SU.
- Uses a standard mapping language (R2RML) to build their KG
- Engineering Value and reproducibility with all the artefacts available

## Weaknesses

- The query comparison between Listing 5 and 6 or Listing 7 and 8 seems unfair. The *perceived* complexity will not be fairly evaluated when the first "conventional" queries are done on a graph modelled for SU. Is it possible that it makes these "conventional" queries appear more complex than on a graph modelled in the "conventional" way?
  - A minor note about the `ORDER BY` present in Listing 5 but not 6\.
  - Additionally, to the juxtaposition of Listings 7 and 8, how I understand them is that all the query results were already constructed during KGC for query 8, thus just needing to select and construct, but for query 7, we need to construct it live, leading to a more complex query.

## Questions

- Could you clarify who are the **users** you are talking about?
  - Currently, I only can imagine these are more domain experts, but the only time we have more information about them is in "even for **users** with a *background in technical fields*" (p. 1\)
  - Then later: "formulate competency questions with **domain experts** that aim to cover a multitude of scenarios of questions users **may** have for the KG" (p. 2\) are these the same people?
  - Then when talking about cognitive interoperability, we have "accessible to database architects, data scientists, and domain experts alike" (p. 2\)
  - Because sometimes I think you talk about different kinds of users, for example, with "Formulating this query requires users to understand and traverse the KG's schema." (p. 10\) then later, "Additionally, these visualizations can be saved as graph snapshots within GraphDB and shared between users. Snapshots could be utilized further in user interfaces to give users a better grasp of KG contents." (p. 12\) Are these the same users? I would not typically expect a domain expert to be able to write SPARQL queries, for instance.
- **Reification, RDF-star, and Singleton Properties**: I am curious about your thoughts on recent developments in **RDF 1.2**. The current justification for using named graphs for Semantic Units is understandable, but RDF 1.2 introduces the ability to natively reify multiple triples (see the example below). Have you considered this approach for the Semantic Units framework? Are there specific contraindications, or is it primarily a matter of the implementation timeline?
- **Competency Questions**: The CQ4 competency question seems more a meta-graph question than a domain question. How is it of interest for domain experts?
- **Lessons Learned**: If possible, could you elaborate on why handwritten mappings for SU are prone to errors and time-consuming? Is this bottleneck due to the general verbosity of authoring R2RML in Turtle, or is it specifically exacerbated by the Semantic Units?
- **Lessons Learned**: Since this work moves beyond small prototypes to a real-world implementation, it would be highly valuable to discuss what you learned regarding scalability. Specifically, how does the Semantic Units concept scale in terms of query execution performance and overall graph size for larger datasets?
- You said you were unable to conduct a *comprehensive* user evaluation, but can you already give a glimpse of it, or is it planned for future work?

## Suggestions

- **Modelling Challenges**: You don't mention how you overcame the modelling challenges, which could have been of interest.
- **Implementation**: The phrase is slightly confusing: "We choose the R2RML mapping language and processing engine". It might read better to distinguish the two by calling the latter an RML processor (specifically **R2RML-F**), and you could cite the tool's article as suggested on their repository.
- **Listing 5:** There is an `ORDER BY` clause present in Listing 5 but not in Listing 6\. If this was not intended, removing it would make the baseline comparison slightly more equitable.
- **Figure 4**: Add a more visual legend with the colours in the diagram itself, or add small colour badges in the text at least.


RDF1.2 example:
```turtle
<< ex:Paper123 ex:acceptedAt conf:ESWC2026 ~ ex:acceptanceRecord456 >> .
<< ex:Paper123 ex:presentedBy ex:Alice ~ ex:acceptanceRecord456 >> .

ex:acceptanceRecord456
	ex:recordedBy ex:ProgramCommittee ;
	ex:date "2026-04-15"^^xsd:date ;
	ex:decision "Accepted" .

# or with an alternative syntaxe

ex:acceptanceRecord456 rdf:reifies
		<<( ex:Paper123 ex:acceptedAt conf:ESWC2026 )>>,
		<<( ex:Paper123 ex:presentedBy ex:Alice )>> .
```

---

### Official Review · ~Davide_Lanti1 · 2026-04-03
**The paper presents a promising and practically relevant direction, but would benefit from clearer definitions and a more systematic evaluation to better understand trade-offs.**

**Rating:** 6
**Confidence:** 4

**Review:**

# Recap

The paper presents a practically motivated and well-implemented approach to structuring knowledge graphs into user-meaningful units, with the aim of supporting better organization, provenance handling, and potential exploration, albeit without strong formal grounding or empirical validation.

# Strengths

- The idea of Semantic Unit (SU) seems very interesting both at the methodological and practical levels, as increasing the user experience when interacting with RDF/OWL Knowledge Bases is a practically relevant and never-solved problem.

- The choice of using named graphs as a means of realizing the SU approach is reasonable and well-justified. The qualitative considerations provided in the Related Work section, when comparing the approach against alternative solutions (e.g., RDF-star) appear sound.

- The approach comes with a concrete implementation and a qualitative evaluation, and the resulting resource is made available through a stable identifier (DOI).

- The evaluation is on a non-trivial, real-world dataset. The mapping effort appears substantial.

- The authors explicitly acknowledge missing user evaluation, tooling gaps, and open questions, which increases the credibility of the contribution.

# Weaknesses

- Several terms require a formal definition: semantically meaningful, smallest unit, statement. Especially the latter: what is a statement? Is it expressed in natural language, FOL, or some specific higher-level formalism? Without information on the granularity of the statement, two different modelers may create completely different units, which would not "talk" to each other. As an example: "A is the first author of P in 2020". What is this? A statement? Two statements? This ambiguity weakens reproducibility and inter-model consistency.

- The notion of the "subject" of a statement is introduced, but its intended semantics and cardinality are underspecified, leaving room for incompatible modeling choices. For instance, in the sentence "Person A is the first author of Publication P", a modeler might decide that A is the subject, while another might decide that P is the subject. Can a statement have multiple subjects?

- The approach seemingly introduces a named graph per statement, along with associated metadata and linking structures. This appears to lead to a rather heavy expansion of metadata: the resulting KG contains roughly 500k triples, but the paper does not isolate how much of this is due to the SU layer.

- As the paper does not contain formal definitions, I also wonder about the actual formal semantics of what is being done. Standard OWL/RDF reasoning does not, by itself, assign special semantics to the SU grouping layer; the main strength of the approach seems to be methodological and query-/validation-oriented, rather than semantic in the logical sense.

- The paper does not evaluate the impact of the additional SU structure on query complexity, maintainability, or runtime.

- It is unclear whether adopting a different approach (e.g., RDF-star, reification patterns) would have simplified the authors' work. Intuitively, I believe the approach has merits, as highlighted by the qualitative considerations the authors provide in the related work section. However, the contribution would have been significantly stronger if the work had included a quantitative evaluation comparing alternative approaches (even on smaller scenarios). This is missing in the current evaluation.

- Qualitatively speaking, if I compare Listings 5 and 6, I understand what is happening in Listing 5 (non-SU approach), but Listing 6 is hard to interpret locally, because the semantics of the specific compound-unit IRI are not made explicit in the query itself, and the role of the transitive path * is not discussed. Since the authors aim to show that Listing 6 is more user-friendly, I would suggest revising the example (which appears feasible).

- The paper claims to evaluate two query approaches, but demonstrates this only for a single competency question (CQ1), in Listings 5 and 6. Listing 7 presents an SU-based query for a more complex case, while Listings 3 and 4 show generic SU query patterns. The remaining competency questions lack corresponding baseline queries, limiting the strength and reproducibility of the comparison. Hence, the claim appears overstated, as no comparison with a "conventional" version is shown for these cases. Having a single comparison for one query (CQ1) does not seem sufficient to support a general comparison of "two query approaches," as stated in the paper.

- The distinction between star-shaped and set-retrieval queries is asserted but not empirically substantiated, as only a single paired example is provided and no systematic comparison or metrics are used.

- The paper shows that the SU organization can support a certain style of visualization, but it does not demonstrate that such visualizations depend on SU modeling, nor that they are better than views derived directly from the base graph.

- Handling 11k lines of R2RML mappings suggests a substantial implementation effort; it is unclear to what extent this complexity stems from the SU approach itself versus the underlying data transformation.

- The paper assumes domain knowledge and does not define "plot" for non-ecology readers; Figure 3 therefore becomes hard to interpret outside the domain. Still with respect to Figure 3, the use of sio:hasSource is conceptually awkward because it appears to express provenance about a plot collection, rather than a domain relation between real-world entities.

# Overall Assessment

The paper presents a promising and practically relevant direction, but would benefit from clearer definitions and a more systematic evaluation to better understand trade-offs.

# Minor

Schematic inconsistencies: based on the description in the paragraph, a more appropriate term is "Schema misalignments".

---

### Official Review · ~Ben_De_Meester1 · 2026-04-04
**Strong Writing and Promising Ideas, but Clearer Framing and Evaluation Needed**

**Rating:** 6
**Confidence:** 3

**Review:**

### High-Level

- The paper is well written, excluding some minor paragraphs that are written too dense.
- SU and all its facets is posed as existing related work, but references are mostly arxiv preprints. I would pay attention to phrasing that you do not mislead the reader, making him/her think this is much more mature work than it actually is (i.e., all claims coming from references 8, 10, 11 are from preprints, I would review the paper as if you would not use those preprints as actual references but just footnotes: are your claims still defensible? I don't feel like all of them are)
- Rename the 'Method' section to, e.g., 'Knowledge Graph Schema' (and then the implementation section can be called 'Knowledge Graph Construction'): the current contents of the Method section is not at all describing a method.
- The base premise/title is a bit misleading: the evaluation is much more an evaluation of the benefits of SU in general, not on the KG Construction part. There is some relevance to the workshop in the description of the advantages of R2RML in this context,but other than that, this paper is much more about 'these are the advantages of using this SU structure, applied to a project', and not really about the KG Construction. Relevance to the workshop is thus, in my opinion, limited.

### Details

#### Introduction

- "Knowledge graphs (KGs) have received ever growing attention in recent decades and are used in many applications in industry and science": 'technology X is used more and more' is not a good reason to use technology X: what are the actually reasons KG are relevant to your context? Given you're in a KG conference/workshop, you can skip that vague general statement, and just immediately start with "Interaction with Knowledge Graphs is notoriously difficult, [...]".
  - In fact, right after this, you actually specify the actual advantages of KGs --> this should be your context
  - I prefer my introduction structure of following flow: what's the context --> what's the problem in this context --> why is this an important problem --> what's our proposal (in principles/high-level argumentation). See the excellent summaries at https://principiae.be/X0100.php
- There's something off in the last sentence of your first paragraph, maybe a misaligned ; vs , ?
  - Also, this is where you make big claims but don't have any references.
- "Semantic Units (SUs) – semantically meaningful, named subgraphs in a knowledge graph – have been proposed in the literature [8, 9, 10, 11] to support users in understanding a KG’s contents" --> well, 3 references are self-published arxiv publications, all 4 are from the same co-author of this paper.. I'd be more accurate "In previous work [9], we introduced Semantic Units to support users in understanding a KG's contents", your current phrasing misleads the reader as if this is established work. No need to bluf, it actually diminishes your credibility as academic
- I don't really like your ending, it's a bit weirdly phrased (2 contributions... first, second, further, finally), halfly introducing the paper sections but not all. I'd make this much more explicit: our 2 contributions are X and Y. The paper is structured as follows: we start with background (section 2), ...

#### Background

- I feel your first paragraph would benefit from explicitly mentioning the number or challenges you face, and then really enumerating them
- the different SU categories are described in a separate preprint, but I would pay attention not to pose this as existing work: this is your previous, unreviewed work you build upon, so this should remain up for debate. The current phrasing kind of misleads the user in my opinion.

#### Related Work

- considering RDF-Star, I miss a comment on how this is currently taken up by the RDF 1.2 WG
- I am very biased (I am co-author of an accepted paper at QKG workshop at ESWC2026, Pieter Colpaert https://pietercolpaert.be/ will also be there, feel free to try and find him to talk to him about this more), but relevant related work is https://knowledgeonwebscale.github.io/rdf-context-associations/ : basically the same issues and proposed solution, with one difference: the use of blank node graphs instead of named graphs, as that allows you the have immutability of the SUs when merging multiple in a triple store.

#### Method

- For the first part, it's not clear to me which of this is new compared to [25]: be explicit in what is novel in this paper, and what is included to create a self-standing paper. Without such explicit mentions, I assume you reuse (so I don't see the KG as a contributions of the paper at hand).
  - I now notice [25] is a master's thesis of the first author, seems fine to include this as a contribution to this paper
  - However, the title is very misleading: the 'Method' section basically introduces the KG schema: what does that have to do with method?

#### Implementation

- Another advantage of using R2RML is the advantage of using a declarative language, and thus a transparent data transformation process, instead of a black-box hardcoded script.
- kudos for publishing all technical artefacts as stable open publications. Really increases the weight of your paper.

#### Evaluation and Discussion

- I do not understand why the relation direction between role and person is different between Listing 5 and Listing 6: is this important? If not, why have that distinction for this example?
- "However, if users execute the graph exploration query in Listing 4 before writing the set retrieval query, they can reuse the list of statement units and their content-graphs." --> Make explicit what you mean by this: SU allows for automatic discovery of relevant properties?
- Listing 7 will have very unclear results, no? If a publication has 2 input datasets and 5 output datasets, the same publication with be returned 7 times, with a lot of duplication. Why did you choose to do this in 1 sparql query instead of multiple?
- Again, [36] and [37] are self-published pre-prints: I suggest to be more explicit about what is your work, and what is ongoing work (36 and 37 are both in my opinion)
- For me, it's weird that Future Work is mentioned here and not in conclusion, but that might be personal preference

#### Conclusion

- conclusion is mostly summarizing what you already wrote, that's a missed opportunity: conclusion can be more of a critical review of your own work (this went well, this is too simple, this has huge potential if people would implement it, etc)
  - future work could move to here

---

### Decision · Program_Chairs · 2026-04-09

**Decision:**

Accept

**Comment:**

This paper has been selected for presentation at the KGC workshop. We strongly encourage the authors to consider the reviews whilst revising the paper. Camera-ready instructions will soon follow.